# Long-term HIV care outcomes under universal HIV treatment guidelines: A retrospective cohort study in 25 countries

Ellen Brazier[1,2]*, Olga Tymejczyk[1], Kara Wools-Kaloustian[3], Awachana Jiamsakul[4], Marco Tulio Luque Torres[5], Jennifer S. Lee[6], Lisa Abuogi[7], Vohith Khol[8], Fernando Mejía Cordero[9], Keri N. Althoff[6], Matthew G. Law[10], Denis Nash[1,2], on behalf of the International epidemiology Databases to Evaluate AIDS (IeDEA)¶

1 City University of New York, Institute for Implementation Science in Population Health (ISPH), New York, New York, United States of America, 2 City University of New York, Graduate School of Public Health and Health Policy, New York, New York, United States of America, 3 Department of Medicine, Indiana University School of Medicine, Indianapolis, Indiana, United States of America, 4 The Kirby Institute, University of New South Wales, Sydney, Australia, 5 Department of Pediatrics, Instituto Hondureño de Seguridad Social and Hospital Escuela Universitario, Tegucigalpa, Honduras, 6 Department of Epidemiology, Johns Hopkins Bloomberg School of Public Health, Baltimore, Maryland, United States of America, 7 Department of Pediatrics, University of Colorado School of Medicine and Children's Hospital Colorado, Aurora, Colorado, United States of America, 8 National Center for HIV/AIDS, Dermatology and STDs, Phnom Penh, Cambodia, 9 Instituto de Medicina Tropical Alexander von Humboldt, Universidad Peruana Cayetano Heredia, Lima, Peru, 10 The Kirby Institute, University of New South Wales, Sydney, Australia

¶ Membership of IeDEA is provided in S1 Text.
* ellen.brazier@sph.cuny.edu

**Data Availability Statement:** Complete data for this study cannot be posted in a supplemental file or a public repository because of legal and ethical restrictions. The principles of collaboration of

## Abstract

### Background

While national adoption of universal HIV treatment guidelines has led to improved, timely uptake of antiretroviral therapy (ART), longer-term care outcomes are understudied. There is little data from real-world service delivery settings on patient attrition, viral load (VL) monitoring, and viral suppression (VS) at 24 and 36 months after HIV treatment initiation.

### Methods and findings

For this retrospective cohort analysis, we used observational data from 25 countries in the International epidemiology Databases to Evaluate AIDS (IeDEA) consortium's Asia-Pacific, Central Africa, East Africa, Central/South America, and North America regions for patients who were ART naïve and aged ≥15 years at care enrollment between 24 months before and 12 months after national adoption of universal treatment guidelines, occurring 2012 to 2018. We estimated crude cumulative incidence of loss-to-clinic (CI-LTC) at 12, 24, and 36 months after enrollment among patients enrolling in care before and after guideline adoption using competing risks regression. Guideline change–associated hazard ratios of LTC at each time point after enrollment were estimated via cause-specific Cox proportional hazards regression models. Modified Poisson regression was used to estimate relative risks of retention, VL monitoring, and VS at 12, 24, and 36 months after ART initiation. There were 66,963 patients enrolling in HIV care at 109 clinics with ≥12 months of follow-up time after

IeDEA and the regulatory requirements of the different IRBs of our participating sites (sometimes representing national IRBs of ministries of health) require the submission of a project concept proposal and approval by the IeDEA Executive Committee. To request data, please review IeDEA guidance available at: https://www.iedea.org/resources/multiregional-research-sops-templates/ and contact the Executive Committee (https://www.iedea.org/working-groups/executive-committee/). Signing of a data sharing agreement may also be required.

**Funding:** The International Epidemiology Databases to Evaluate AIDS (IeDEA) is supported by the U.S. National Institutes of Health's National Institute of Allergy and Infectious Diseases (https://www.niaid.nih.gov), the Eunice Kennedy Shriver National Institute of Child Health and Human Development (https://www.nichd.nih.gov), the National Cancer Institute (https://www.cancer.gov), the National Institute of Mental Health (https://www.nimh.nih.gov), the National Institute on Drug Abuse (https://nida.nih.gov), the National Heart, Lung, and Blood Institute (https://www.nhlbi.nih.gov), the National Institute on Alcohol Abuse and Alcoholism (https://www.niaaa.nih.gov), the National Institute of Diabetes and Digestive and Kidney Diseases (https://www.niddk.nih.gov), the Fogarty International Center (https://www.fic.nih.gov), and the National Library of Medicine (https://www.nlm.nih.gov): Asia-Pacific, U01AI069907 (AJ, MGL and VK); Caribbean, Central and South America network for HIV epidemiology (CCASAnet), U01AI069923 (MTLT and FMC); Central Africa, U01AI096299 (EB, DN and OT); East Africa, U01AI069911 (KWK and LA); NA-ACCORD, U01AI069918 (KNA and JSL). Informatics resources are supported by the Harmonist project, R24AI124872. The funders had no role in study design, data collection and analysis, decision to publish, or preparation of the manuscript.

**Competing interests:** KNA receives royalties from Coursera. She was a consultant to the All of Us Research Program. She serves on the scientific advisory board for TrioHealth Inc. DN received consulting fees from Abbvie and Gilead and is the PI of a research grant from Pfizer to his institution'. All other authors have declared that no competing interests exist.

**Abbreviations:** aHR, adjusted hazard ratio; aRR, adjusted relative risk; ART, antiretroviral therapy; CI-LTC, cumulative incidence of loss-to-clinic; COVID-19, Coronavirus Disease 2019; GEE, generalized estimating equation; IeDEA, International epidemiology Databases to Evaluate AIDS; IQR, interquartile range; NNRTI, non-

enrollment (46,484 [69.4%] enrolling before guideline adoption and 20,479 [30.6%] enrolling afterwards). More than half (54.9%) were females, and median age was 34 years (interquartile range [IQR]: 27 to 43). Mean follow-up time was 51 months (standard deviation: 17 months; range: 12, 110 months). Among patients enrolling before guideline adoption, crude CI-LTC was 23.8% (95% confidence interval [95% CI] 23.4, 24.2) at 12 months, 31.0% (95% CI [30.6, 31.5]) at 24 months, and 37.2% (95% [CI 36.8, 37.7]) at 36 months after enrollment. Adjusting for sex, age group, enrollment CD4, clinic location and type, and country income level, enrolling in care and initiating ART after guideline adoption was associated with increased hazard of LTC at 12 months (adjusted hazard ratio [aHR] 1.25 [95% CI 1.08, 1.44]; $p = 0.003$); 24 months (aHR 1.38 [95% CI 1.19, 1.59]; $p < .001$); and 36 months (aHR 1.34 [95% CI 1.18, 1.53], $p < .001$) compared with enrollment before guideline adoption, with no before–after differences among patients with no record of ART initiation by end of follow-up. Among patients retained after ART initiation, VL monitoring was low, with marginal improvements associated with guideline adoption only at 12 months after ART initiation. Among those with VL monitoring, VS was high at each time point among patients enrolling before guideline adoption (86.0% to 88.8%) and afterwards (86.2% to 90.3%), with no substantive difference associated with guideline adoption. Study limitations include lags in and potential underascertainment of care outcomes in real-world service delivery data and potential lack of generalizability beyond IeDEA sites and regions included in this analysis.

## Conclusions

In this study, adoption of universal HIV treatment guidelines was associated with lower retention after ART initiation out to 36 months of follow-up, with little change in VL monitoring or VS among retained patients. Monitoring long-term HIV care outcomes remains critical to identify and address causes of attrition and gaps in HIV care quality.

## Author summary

### Why was this study done?

- Although universal HIV treatment recommendations have been adopted in national HIV treatment guidelines, longer-term HIV care outcomes under such guidelines are poorly documented and largely limited to single-country studies with short follow-up times.

- No multicountry studies using real-world service delivery data have examined long-term HIV care outcomes associated under universal HIV treatment guidelines.

### What did the researchers do and find?

- With data on 66,963 patients enrolling in HIV care at 109 clinics participating in the International epidemiology Databases to Evaluate AIDS (IeDEA) research consortium across 25 countries where universal HIV treatment guidelines were adopted, we

nucleoside reverse transcriptase inhibitor; PLWH, people living with HIV; VL, viral load; VS, viral suppression; WHO, World Health Organization; 95% CI, 95% confidence interval.

estimated the hazard ratios of loss-to-clinic (LTC) at 12, 24, and 36 months after enrollment, comparing those enrolling in HIV care after guideline adoption to those enrolling before guideline adoption.

- Among 57,615 patients with documented initiation of antiretroviral therapy (ART), we also estimated the relative risks of clinic retention, viral load (VL) monitoring, and viral suppression (VS) at 12, 24, and 36 months after ART initiation, comparing those enrolling after versus before national adoption of universal treatment guidelines.

- Compared with patients enrolling in HIV care and initiating HIV treatment before national adoption of universal treatment guidelines, those enrolling and initiating treatment after guideline adoption had higher risk of being LTC at 12 months, 24 months, and 36 months after enrollment.

- Among patients retained in care after ART initiation, those enrolling in HIV care after the adoption of universal HIV treatment guidelines were more likely to have VL monitoring at 12 months after ART initiation and less likely at 36 months, with no difference at 24 months.

- VS was high at each time point among patients enrolling before and after the adoption of universal HIV treatment guidelines, with no substantive change associated with guideline adoption.

### What do these findings mean?

- Our results raise concerns about long-term retention of patients after ART initiation, as well as the capacity of HIV programs to provide essential aspects of HIV care, including annual VL monitoring for timely identification of adherence problems and treatment failure.

- Our findings that patient retention in care at the clinic where ART was initiated decreased after the adoption of universal HIV treatment guidelines and that there has been no improvement in annual VL monitoring among patients retained in care should motivate efforts to identify and address factors associated with attrition among patients enrolling in HIV care, as well as barriers to routine VL testing in the era of universal treatment of all people living with HIV.

- Study limitations include potential underascertainment of patient outcomes in real-world service delivery data, lags in the availability of real-world service delivery data, and the nonrepresentativeness of the clinics and countries reflected in IeDEA datasets available for analysis.

## Introduction

The World Health Organization (WHO)'s 2015 recommendation for universal treatment for all people living with HIV (PLWH) [1]—known as "Treat-All"—eliminated an important barrier to initiation of antiretroviral therapy (ART) in many settings [2]. While a few high-income countries had universal HIV treatment guidelines in place prior to WHO's 2015 recommendation, most countries around the globe adopted expanded treatment guidelines subsequently,

with an estimated 70% of low- and middle-income countries adopting universal HIV treatment guidelines by the end of 2017 [3].

Observational studies have shown that national adoption of universal treatment guidelines has led to greater uptake and more rapid initiation of ART across diverse country settings [4,5]. Improved treatment uptake and more timely initiation of ART are promising for the reduction of HIV morbidity and mortality in patients, as well as preventing onward transmission of the virus (i.e., treatment as prevention) [6–8]. However, longer-term HIV care outcomes, such as retention in care and timely and sustained viral suppression (VS), under universal treatment guidelines are underresearched and largely limited to small single-country studies with short follow-up times of 6 to 12 months [9–14]. While a community cluster-randomized controlled trial in Uganda and step-wedged randomized trial in Eswatini have reported higher 12-month retention and combined 12-month retention/VS rates among patients initiating treatment under universal treatment guidelines, compared with standard initiation practices [15,16], several observational studies have reported no improvement in care retention—or lower retention—among patients initiating treatment in the era of universal treatment [10–13].

Using data from the International epidemiology Databases to Evaluate AIDS (IeDEA) research consortium, we aimed to estimate loss-to-clinic (LTC) at 12, 24, and 36 months after enrollment in HIV care, comparing those enrolling before and after country-level adoption of universal HIV treatment guidelines. Additionally, we aimed to estimate clinic retention, viral load (VL) testing, and VS at 12, 24, and 36 months after ART initiation.

## Methods

### Data sources

The IeDEA consortium pools observational clinical data on more than 2 million PLWH ever enrolling in HIV care at approximately 400 care and treatment sites in 44 countries [17]. Our study population was drawn from IeDEA's cohorts in the Asia-Pacific, Central Africa, East Africa, the Caribbean, Central and South America, and North America regions, which agreed to the use of their data for this retrospective cohort study, based on a concept proposal approved by IeDEA's executive committee. Deidentified patient data from participating IeDEA cohorts were standardized in accordance with IeDEA data definitions [18]. The research was approved by the City University of New York (CUNY) University Institutional Review Board (#2018–0809).

For each country in participating regional cohorts, we identified the date universal ART eligibility was extended to all adult patients, based on policy documents, literature, and inputs from in-country experts, as described elsewhere [19]. Patients were eligible if they were at least 15 years of age and ART naïve at the time of enrollment in HIV care at an IeDEA site, enrolled in care in the 24 months immediately before national adoption of universal treatment guidelines or in the first 12 months thereafter, and enrolled in care at least 12 months before database closure (i.e., submission of data to IeDEA's regional data centers for processing in accordance with IeDEA's data exchange standards [18]). For cohorts that deidentify patient data by shifting patient encounter dates by <30 days, we excluded all patients enrolling in care within +/− 30 days of the date of guideline adoption. Cohorts where all patient enrollment dates are shifted to a midyear date (i.e., July 1) were excluded, along with clinics where no patients had any records of VL monitoring. Additionally, we excluded patients with missing data for sex or age at enrollment in HIV care, and missing date of death if recorded as deceased. The years of data used in this study ranged from 2010 to 2021.

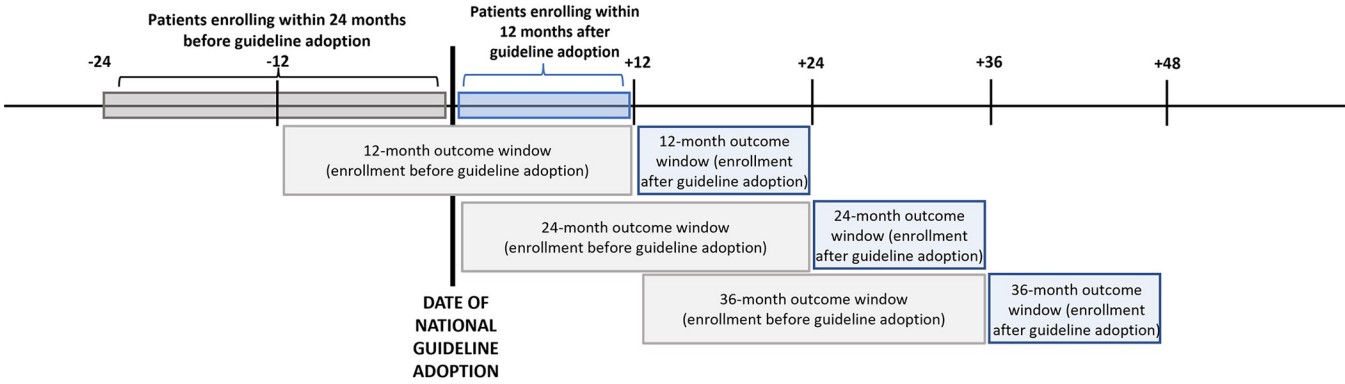

**Fig 1. Study populations and windows for primary outcome ascertainment.**

## Exposure

The exposure of interest was enrollment in HIV care before or after the official date of national adoption of universal HIV treatment guidelines, which, depending on the country, occurred between 2012 and 2018.

## Outcomes

Among all patients who were ART naïve, who enrolled in HIV care in the 24 months before or 12 months after national adoption of universal HIV treatment guidelines, and who had sufficient follow-up time between enrollment and database closure, our primary outcome of interest was LTC by 12, 24, and 36 months after enrollment (Fig 1). Across all regional cohorts, LTC was defined as no evidence of contact (e.g., visits, laboratory testing, or medication pickups) with the clinic of enrollment for at least 12 months prior to database closure [20]. The date of LTC was set at 90 days after the last clinic contact, with patients considered LTC if their date of LTC was before censoring at 12, 24, and 36 months, respectively. Patients not documented as having died or transferred and not classified as LTC by the censoring date were considered retained in care at the clinic.

Among the subset of patients with evidence of ART initiation and sufficient follow-up time between ART initiation and database closure, we examined clinic retention at 12, 24, and 36 months after ART initiation, with retention defined as no documentation of death or transfer to another site of care, and not LTC (as defined above). Among patients on ART and retained in care, we examined VL monitoring, defined as any VL test at 12, 24, and 36 months (+/−3 months) after ART initiation, and among patients with VL monitoring at these time points, we examined VS, defined as VL <1,000 copies/mL. ART, antiretroviral therapy; LTC, loss-to-clinic; VL, viral load; VS, viral suppression.

## Covariates

Patient-level characteristics included sex (male or female); age at enrollment in HIV care (categorized as 15 to 19 years, 20 to 24 years, 25 to 34 years, and >34 years; CD4 count within 90 days (+/−) of enrollment and no more than 30 days after ART initiation (categorized as: ≤200 cells/μL; 201 to 350 cells/μL; 351 to 500 cells/μL; >500 cells/μL; or unknown/missing); ART initiation was defined as the start of a combination antiretroviral treatment regimen before censoring at 12, 24, and 36 months after enrollment, and initial ART regimens were categorized as non-nucleoside reverse transcriptase inhibitor (NNRTI)-based regimens, protease inhibitor–based regimens, integrase inhibitor–based regimens, other/unknown regimens, or none).

Clinic-level characteristics included location (rural/mostly rural versus urban/mostly urban); facility type (i.e., health center, district hospital, regional/university referral hospital, or other); and country income level in 2018 (low, lower-middle, upper-middle, and high income) as reflected in World Bank databases [21].

## Statistical analysis

We used descriptive statistics to compare the characteristics of patients enrolling before and after adoption of universal HIV treatment guidelines, along with LTC at each time point after enrollment, and retention, VL monitoring, and VS after ART initiation. We also described the number and proportion of patients recorded as having transferred or died by each time point after enrollment.

We estimated the crude cumulative incidence (i.e., risk) of LTC at 12, 24, and 36 months after enrollment, stratified by the timing of enrollment relative to the date of guideline adoption and ART initiation status by the censoring time point (i.e., on ART or not by 12, 24, and 36 months after enrollment) via competing risks regression using the Aalen–Johansen estimator [22]. Multivariable cause-specific Cox proportional hazards regression [23] was used to estimate the association between universal treatment guideline adoption and LTC at 12, 24, and 36 months after enrollment, adjusting for the above covariates, and the clustering of patients within clinics was accounted for when fitting the Cox proportional hazards models through the use of a robust sandwich estimator for the covariance matrix. We considered death as a competing risk for LTC, with transfer treated as a censoring variable. We tested for statistical interactions between enrollment period (before versus after guideline adoption) and ART initiation status by censoring time points and stratified results by ART initiation status (i.e., on ART or not on ART).

Among the subset of patients with evidence of ART initiation, we estimated relative risks of the binary outcomes of clinic retention, VL monitoring, and VS at 12, 24, and 36 months after ART initiation via modified Poisson regression models, comparing those enrolling in care after versus before universal HIV treatment adoption. Multivariable models were adjusted for patient and clinic characteristics (sex, age group, enrollment CD4, initial regimen type, clinic location, facility type, and country income level), using generalized estimating equations (GEEs) to account for clustering within clinics.

In a sensitivity analysis, we compared relative hazards of LTC at 12 months after enrollment and relative risks of care retention, VL monitoring, and VS at 12 months after ART initiation among patients enrolling in HIV care in the 12 months after adoption of universal HIV treatment guidelines versus the 13 to 24 months before guideline adoption (i.e., excluding patients enrolling in care during the 12 months immediately before guideline adoption whose outcome ascertainment window was entirely in the period after guideline adoption) (Fig 2).

An initial concept proposal for this analysis (S1 Concept proposal) outlined exposures, types of outcomes of interest, participating IeDEA cohorts, and general analytic approach and was approved by the IeDEA Executive Committee in August 2018. The use of competing risks and cause-specific hazards regression to examine LTC was not prespecified, with these methods introduced a priori to avoid potential bias that might result from failure to account for competing events in outcome estimation. In a second sensitivity analysis, performed in response to reviewer feedback, we excluded countries where universal treatment guidelines were adopted in 2017 and 2018, to assess whether our estimates of HIV care outcomes at 24 and 36 months could have been affected by the Coronavirus Disease 2019 (COVID-19) pandemic in countries with late adoption of universal HIV treatment guidelines.

All statistical analyses were performed using SAS 9.4 (SAS Institute, Cary, NC). A *p*-value less than 0.05 was considered statistically significant. This study is reported as per the

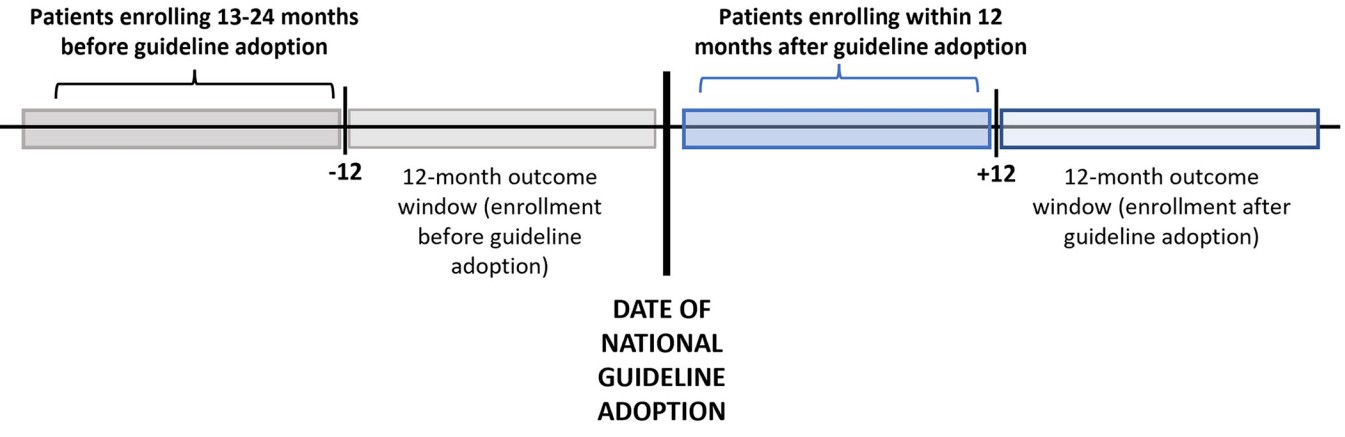

**Fig 2. Sensitivity analysis: Study populations and windows for outcome ascertainment.**

Strengthening the Reporting of Observational Studies in Epidemiology (STROBE) guideline (S1 Checklist).

## Results

Of 132,776 patients in the initial data submission from participating IeDEA regions, 54,961 did not meet eligibility criteria related to age, ART-naïve status, or timing of enrollment, and 10,689 were excluded because they were in care at clinics with no VL testing or in cohorts where patient data are anonymized by shifting dates of care (Fig 3). Additionally, 163 patients were excluded because of missing data related to age, sex, or death date. After these exclusions, there were 66,963 patients who were ART naïve and aged ≥15 years at enrollment in HIV care at 109 clinics in 25 countries (Argentina, Australia, Brazil, Burundi, Cambodia, Cameroon, Canada, Chile, Congo, Democratic Republic of Congo, Haiti, Honduras, Hong Kong, Japan, Kenya, Malaysia, Mexico, Peru, Rwanda, South Korea, Tanzania, Thailand, Uganda, United States, and Vietnam). The sample included 46,484 (69.4%) who enrolled in the 24 months prior to national adoption of universal HIV treatment guidelines and 20,479 (30.6%) who enrolled in the 12 months after guideline adoption (Table 1). Mean follow-up time was 51 months (standard deviation: 17 months; range: 12, 110 months).

Among the full sample of patients with at least 12 months of follow-up time between enrollment and database closure, 54.9% were female, and the median age was 34 years (interquartile range [IQR]: 27, 43), with little difference by period of enrollment (before vs. after adoption of universal treatment guidelines). A smaller proportion of patients enrolling in care after adoption of universal treatment guidelines had CD4 testing results recorded at enrollment (33.8% after vs. 47.4% before; $p < 0.001$); however, among those with any enrollment CD4 test, median CD4 counts were clinically similar among those enrolling before and after adoption of universal treatment guidelines (302 cells/μL [IQR: 136, 492] vs. 315 cells/μL [IQR: 138, 518]). The proportion of patients who had initiated ART by 12 months after enrollment was higher among those enrolling in the year after guideline adoption, compared with those enrolling before (87.6% vs. 78.4%; $p < 0.001$), and among those initiating ART within 12 months of enrollment, the median time from enrollment to treatment initiation decreased from 14 days (IQR: 0, 44) to 0 days (IQR: 0, 14).

The majority of patients (78.8%) were in care at clinics in urban/mostly urban settings, and almost half (45.3%) were at tertiary hospitals, with 24.4% at health centers and 24.7% at district hospitals. Most patients (83.2%) were from low- and lower-middle-income countries and

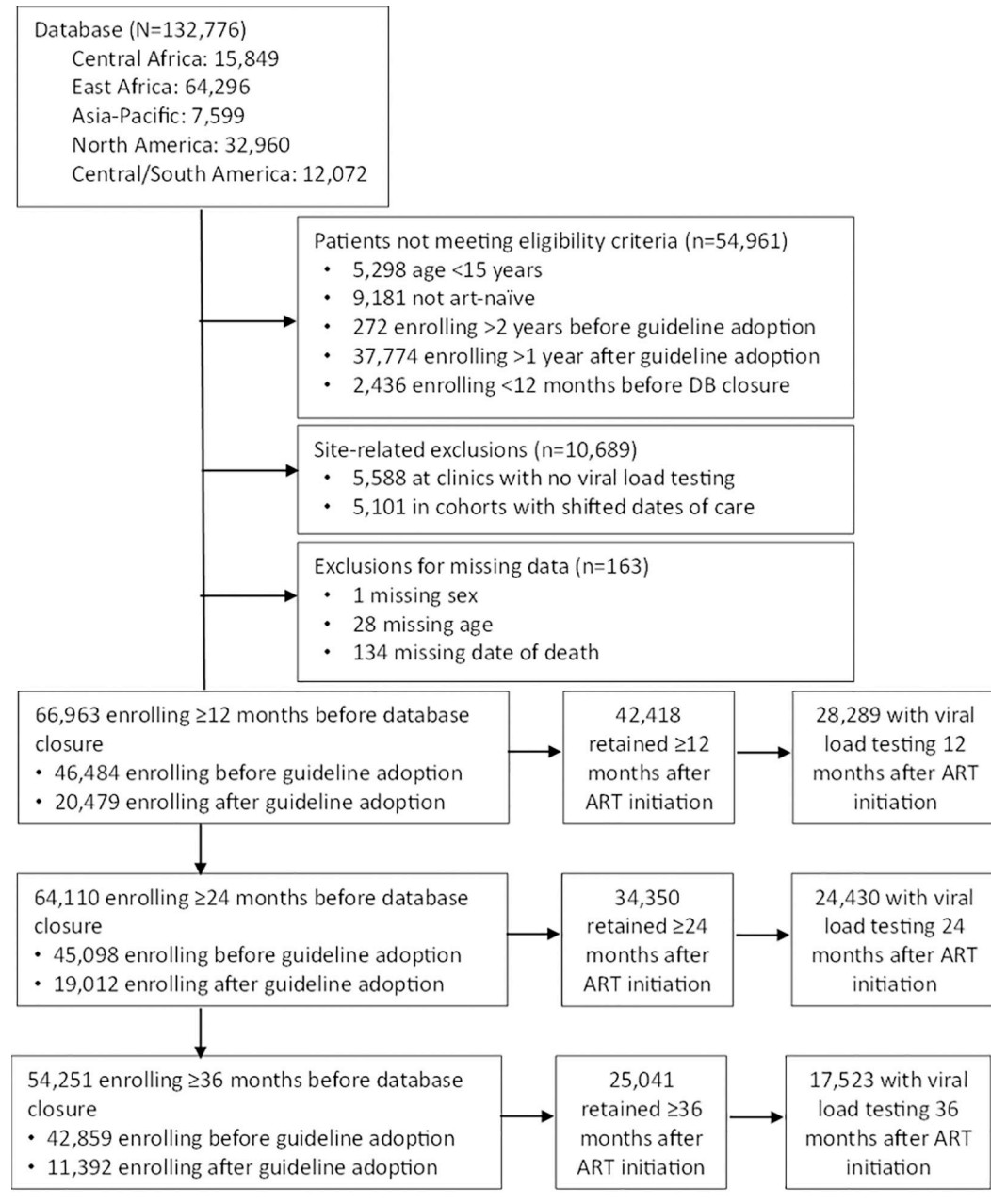

**Fig 3. Sample flow diagram.** ART, antiretroviral therapy.

countries that adopted universal HIV treatment guidelines in 2016 (75.4%), with small minori-ties of patients in countries with earlier (2012 to 2015) or later adoption of such guidelines (2017 to 2018).

Of patients enrolling before national adoption of universal HIV treatment guidelines, 45,098 (97.0%) had at least 24 months of follow-up time before database closure, and 42,859 (92.2%) had at least 36 months of follow-up time. Among patients enrolling after guideline adoption, 19,012 (92.8%) had at least 24 months of follow-up time and 11,392 (55.6%) had at least 36 months of follow-up time. The distribution of patient and clinic characteristics among those with 24 and 36 months of follow-up time after enrollment was similar to those of patients

**Table 1. Baseline characteristics among patients with at least 12 months of potential follow-up time after enrollment, by period of enrollment (before vs. after adoption of universal HIV treatment guidelines).**

| Patient characteristics | N (%) | Before guideline adoption | After guideline adoption | p-Value[a] |
|---|---|---|---|---|
| All patients | 66,963 | 46,484 (69.4) | 20,479 (30.6) | |
| Sex | | | | |
| Male | 30,233 (45.1) | 20,872 (44.9) | 9,361 (45.7) | 0.053 |
| Female | 36,730 (54.9) | 25,612 (55.1) | 11,118 (54.3) | |
| Age (in years) | | | | |
| Median age (IQR) | 34 (27, 43) | 34 (27, 43) | 34 (27, 43) | 0.006[b] |
| 15–19 years | 2,939 (4.4) | 2,085 (4.5) | 854 (4.2) | 0.046 |
| 20–24 years | 9,203 (13.7) | 6,469 (13.9) | 2,734 (13.4) | |
| 25–34 years | 21,729 (32.4) | 15,050 (32.4) | 6,679 (32.6) | |
| >34 years | 33,092 (49.4) | 22,880 (49.2) | 10,212 (49.9) | |
| CD4 count at enrollment | | | | |
| No CD4 count at enrollment | 38,009 (56.8) | 24,455 (52.6) | 13,554 (66.2) | <0.001 |
| Any CD4 count at enrollment | 28,954 (43.2) | 22,029 (47.4) | 6,925 (33.8) | |
| Median CD4 count (IQR) | 304 (136, 497) | 302 (136, 492) | 315 (138, 518) | <0.001[b] |
| <200 cells/μl | 9,972 (34.4) | 7,637 (34.7) | 2,335 (33.7) | <0.001 |
| 200–349 cells/μl | 6,492 (22.4) | 5,005 (22.7) | 1,487 (21.5) | |
| 350–499 cells/μl | 5,359 (18.5) | 4,104 (18.6) | 1,255 (18.1) | |
| > = 500 cells/μl | 7,131 (24.6) | 5,283 (24.0) | 1,848 (26.7) | |
| Initiation of ART by censoring endpoint (12 months) | | | | |
| Not on ART | 12,575 (18.8) | 10,041 (21.6) | 2,534 (12.4) | <0.001 |
| On ART | 54,388 (81.2) | 36,443 (78.4) | 17,945 (87.6) | |
| Mean time in days (SD) to ART initiation among patients initiating ART within 12 months of enrollment | 33.5 (64.3) | 42.3 (71.2) | 15.7 (42.0) | <0.001[c] |
| Median time in days (IQR) to ART initiation among patients initiating ART within 12 months of enrollment | 7 (0, 32) | 14 (0, 44) | 0 (0, 14) | <0.001[b] |
| *Clinic characteristics* | | | | |
| Location | | | | |
| Urban/mostly urban | 52,737 (78.8) | 36,467 (78.5) | 16,270 (79.4) | 0.004 |
| Rural/mostly rural | 14,226 (21.2) | 10,017 (21.5) | 4,209 (20.6) | |
| Facility type | | | | |
| Health center | 16,321 (24.4) | 11,152 (24.0) | 5,169 (25.2) | <0.001 |
| District hospital | 16,559 (24.7) | 11,618 (25.0) | 4,941 (24.1) | |
| Regional, provincial or university hospital | 30,301 (45.3) | 20,941 (45.0) | 9,360 (45.7) | |
| Other[d] | 3,782 (5.6) | 2,773 (6.0) | 1,009 (4.9) | |
| Country income level (2018) | | | | |
| Low income | 24,469 (36.5) | 16,600 (35.7) | 7,869 (38.4) | <0.001 |
| Lower-middle income | 31,277 (46.7) | 21,887 (47.1) | 9,390 (45.9) | |
| Upper-middle income | 2,419 (3.6) | 1,725 (3.7) | 694 (3.4) | |
| High income | 8,798 (13.1) | 6,272 (13.5) | 2,526 (12.3) | |
| Geographic region | | | | |
| Asia-Pacific | 952 (1.4%) | 580 (1.2%) | 372 (1.8%) | <0.001 |
| Central/South America | 9,421 (14.1%) | 6,335 (13.6%) | 3,086 (15.1%) | |
| Central Africa | 7,019 (10.5%) | 4,946 (10.6%) | 2,073 (10.1%) | |
| East Africa | 42,093 (62.9%) | 29,354 (63.1%) | 12,739 (62.2%) | |
| North America | 7,478 (11.2%) | 5,269 (11.3%) | 2,209 (10.8%) | |
| Year of national adoption of universal HIV treatment guidelines | | | | |

(*Continued*)

**Table 1.** (Continued)

| Patient characteristics | N (%) | Before guideline adoption | After guideline adoption | p-Value[a] |
|---|---|---|---|---|
| 2012–2015[e] | 8,633 (12.9) | 6,091 (13.1) | 2,542 (12.4) | <0.001 |
| 2016[f] | 50,471 (75.4) | 34,705 (74.7) | 15,766 (77.0) | |
| 2017–2018[g] | 7,859 (11.7) | 5,688 (12.2) | 2,171 (10.6) | |

ART, antiretroviral therapy; IQR, interquartile range; SD, standard deviation.

[a]Chi-squared test.

[b]Kruskal–Wallis test.

[c]t Test (Satterthwaite).

[d]Other facility types are sites that report data as part of a network comprising clinics and hospitals.

[e]Argentina, Australia, Brazil, Canada, South Korea, Mexico, Thailand, United States.

[f]Burundi, Cambodia, Cameroon, Haiti, Hong Kong, Japan, Kenya, Rwanda, Uganda.

[g]Chile, Congo, Democratic Republic of Congo, Honduras, Malaysia, Peru, Tanzania, Vietnam.

with 12 months of follow-up time, with few appreciable differences between patients enrolling after versus before adoption of universal treatment guidelines (S1 Table).

Among all patients with at least 12 months of follow-up time after enrollment, 69.2% were retained in care at 12 months after enrollment, 24.2% were LTC, 3.3% were documented as deceased, and 3.4% were documented transfers (Table 2). Compared with patients enrolling before adoption of universal treatment guidelines, higher proportions of patients enrolling after guideline adoption were recorded as transfers at 12 months (4.1% versus 3.0%; $p < 0.001$), 24 months (6.0% versus 4.5%; $p < 0.001$), and 36 months after enrollment (6.7% versus 5.6%; $p < 0.001$) or were LTC at each time point (12 months: 25.1% versus 23.8%, $p < 0.001$; 24 months: 35.9% versus 31.0%, $p < 0.001$; 36 months: 40.2% versus 37.2%, $p < 0.001$). Among all patients with a record of ART initiation, 73.6% were retained in care at 12 months after ART initiation, with 62.0% and 53.4%, respectively, retained at 24 and 36 months after ART initiation. Among patients on ART, clinic retention was higher at each time point among those enrolling in care before guideline adoption than after (75.1% versus 70.5% at 12 months, $p < 0.001$; 64.8% versus 55.8% at 24 months, $p < 0.001$; and 55.0% versus 48.2% at 36 months, $p < 0.001$).

Among all patients retained in care after ART initiation, VL monitoring ranged from 66.7% at 12 months after ART initiation to 71.1% at 24 months and 70.0% at 36 months (Table 2), with higher proportions of patients enrolling after adoption of universal treatment guidelines having VL monitoring at 12 and 24 months (73.2% and 73.4%, respectively), compared with before (63.8% and 70.3%). In contrast, VL monitoring at 36 months was lower among patients enrolling after guideline adoption (63.7%) than before (71.6%, $p < 0.001$). Among patients retained after ART initiation with VL test results, VS ranged from 86.3% at 12 months to 89.1% at 36 months, with marginal increases in VS at 24 and 36 months among patients enrolling after adoption of universal HIV treatment guidelines (88.8% and 90.3%, respectively) than before (87.2% and 88.8%, respectively).

The crude cumulative incidence of LTC (CI-LTC) at each time point is shown in Fig 4, stratified by timing of enrollment relative to national adoption of universal treatment guidelines. At each time point, CI-LTC was higher among those enrolling after guideline adoption than before, with insubstantial differences at 12 months after enrollment (CI-LTC: 25.1% versus 23.8%), and larger differences at 24 months (CI-LTC: 35.9% versus 31.0%) and 36 months

**Table 2. HIV care outcomes by 12, 24, and 36 months after enrollment, overall and by ART initiation status by censoring time point.**

| Care outcome | Cohort (N) | Among all patients in cohort | Before guideline adoption | After guideline adoption | p-Value[a] |
|---|---|---|---|---|---|
| | | n (%) | n (%) | n (%) | |
| **Death** | | | | | |
| 12 months after enrollment | 66,963 | 2,185 (3.3) | 1,503 (3.2%) | 682 (3.3%) | 0.516 |
| 24 months after enrollment | 64,110 | 2,623 (4.1) | 1,817 (4.0%) | 806 (4.2%) | 0.219 |
| 36 months after enrollment | 54,251 | 2,471 (4.6) | 1,978 (4.6%) | 493 (4.3%) | 0.191 |
| **Transfer** | | | | | |
| 12 months after enrollment | 66,963 | 2,249 (3.4) | 1,404 (3.0%) | 845 (4.1%) | <0.001 |
| 24 months after enrollment | 64,110 | 3,177 (5.0) | 2,045 (4.5%) | 1,132 (6.0%) | <0.001 |
| 36 months after enrollment | 54,251 | 3,147 (5.8) | 2,380 (5.6%) | 767 (6.7%) | <0.001 |
| **LTC** | | | | | |
| 12 months after enrollment | 66,963 | 16,191 (24.2) | 11,056 (23.8%) | 5,135 (25.1%) | <0.001 |
| 24 months after enrollment | 64,110 | 20,820 (32.5) | 13,997 (31.0%) | 6,823 (35.9%) | <0.001 |
| 36 months after enrollment | 54,251 | 20,526 (37.8) | 15,946 (37.2%) | 4,580 (40.2%) | <0.001 |
| **Retention in care** | | | | | |
| 12 months after enrollment | 66,963 | 46,338 (69.2) | 32,521 (70.0%) | 13,817 (67.5%) | <0.001 |
| 24 months after enrollment | 64,110 | 37,490 (58.5) | 27,239 (60.4%) | 10,251 (53.9%) | <0.001 |
| 36 months after enrollment | 54,251 | 28,107 (51.8) | 22,555 (52.6%) | 5,552 (48.7%) | <0.001 |
| **Retention in care** | | | | | |
| 12 months after ART initiation | 57,615 | 42,418 (73.6) | 29,483 (75.1%) | 12,935 (70.5%) | <0.001 |
| 24 months after ART initiation | 55,416 | 34,350 (62.0) | 24,785 (64.8%) | 9,565 (55.8%) | <0.001 |
| 36 months after ART initiation | 46,850 | 25,041 (53.4) | 20,025 (55.0%) | 5,016 (48.2%) | <0.001 |
| **VL testing among patients initiating ART and retained in care** | | | | | |
| 12 months after ART initiation | 42,418 | 28,289 (66.7) | 18,804 (63.8%) | 9,485 (73.3%) | <0.001 |
| 24 months after ART initiation | 34,350 | 24,430 (71.1) | 17,413 (70.3%) | 7,017 (73.4%) | <0.001 |
| 36 months after ART initiation | 25,041 | 17,523 (70.0) | 14,328 (71.6%) | 3,195 (63.7%) | <0.001 |
| **VS among retained patients with VL test** | | | | | |
| 12 months after ART initiation | 28,289 | 24,404 (86.3) | 16,210 (86.2%) | 8,194 (86.4%) | 0.671 |
| 24 months after ART initiation | 24,430 | 21,413 (87.7) | 15,184 (87.2%) | 6,229 (88.8%) | 0.001 |
| 36 months after ART initiation | 17,523 | 15,606 (89.1) | 12,721 (88.8%) | 2,885 (90.3%) | 0.013 |

ART, antiretroviral therapy; LTC, lost to clinic; VL, viral load; VS, viral suppression.

[a]Chi-squared test.

(CI-LTC: 40.2% versus 37.2%). CI-LTC stratified by ART initiation status is shown in Fig 5. Among patients enrolling before adoption of universal treatment guidelines, CI-LTC at each time point was more than twice as high among patients not on ART, compared with those initiated on ART, with smaller differences by ART initiation status among those enrolling after guideline adoption. Among patients already on ART, CI-LTC at each time point was substantially higher among those enrolling after the adoption of universal treatment guidelines than before; among those not yet on ART, CI-LTC at 24 and 36 months did not differ by period of enrollment (before versus after guideline adoption).

Crude and adjusted hazard ratios (aHRs) of LTC at each time point after enrollment are shown in Table 3, for all patients and stratified by ART initiation status. In the full sample, hazards of LTC at each time point were higher among patients enrolling after adoption of universal treatment guidelines compared with before, but differences were small and not statistically significant. However, the association of universal treatment guidelines with LTC hazards at 24

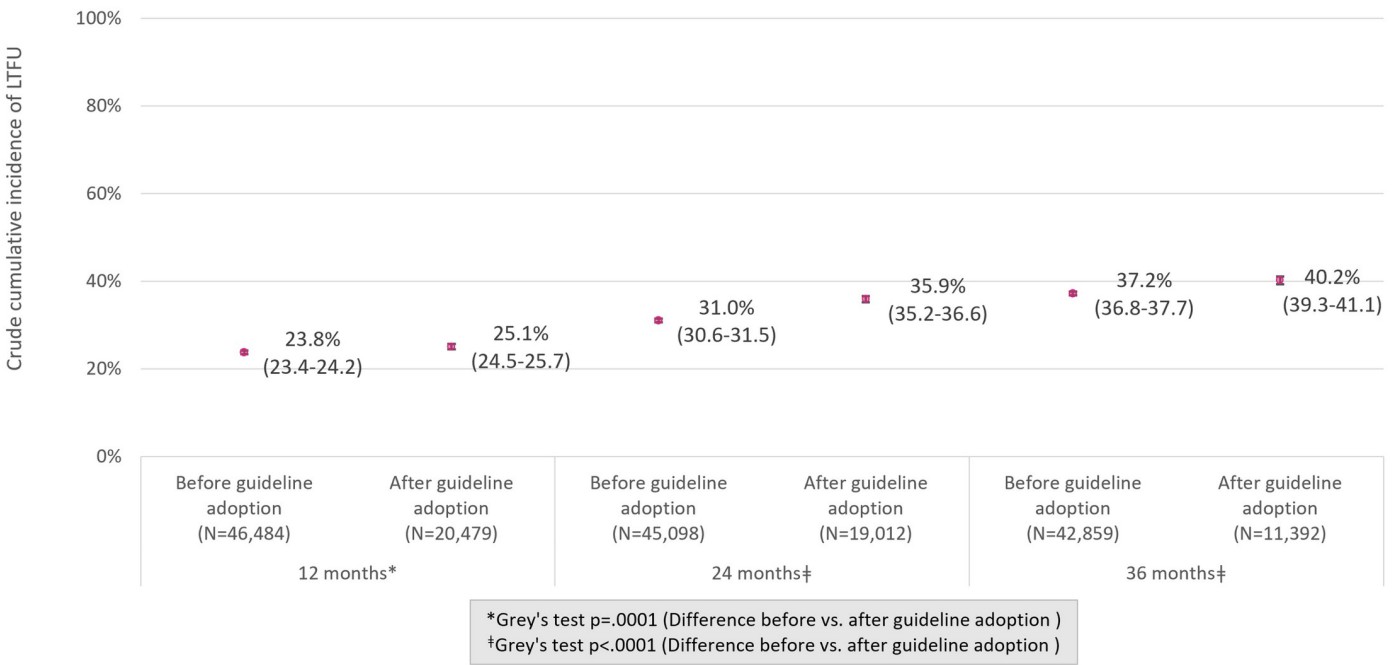

**Fig 4. Crude CI (95% CIs) of LTC at 12, 24, and 36 months after enrollment before vs. after adoption of universal HIV treatment guidelines.** ART, antiretroviral therapy; CI, cumulative incidence; LTC, lost to clinic; 95% CI, 95% confidence interval.

and 36 months after enrollment differed by ART initiation status. Among patients on ART by 24 and 36 months after enrollment, hazards of LTC were substantially higher among patient enrolling after adoption of universal treatment guidelines (24-month aHR 1.38 [95% CI 1.19, 1.59]; $p < 0.001$ and 36-month aHR 1.34 [95% CI: 1.18, 1.53]; $p < 0.001$). In contrast, among

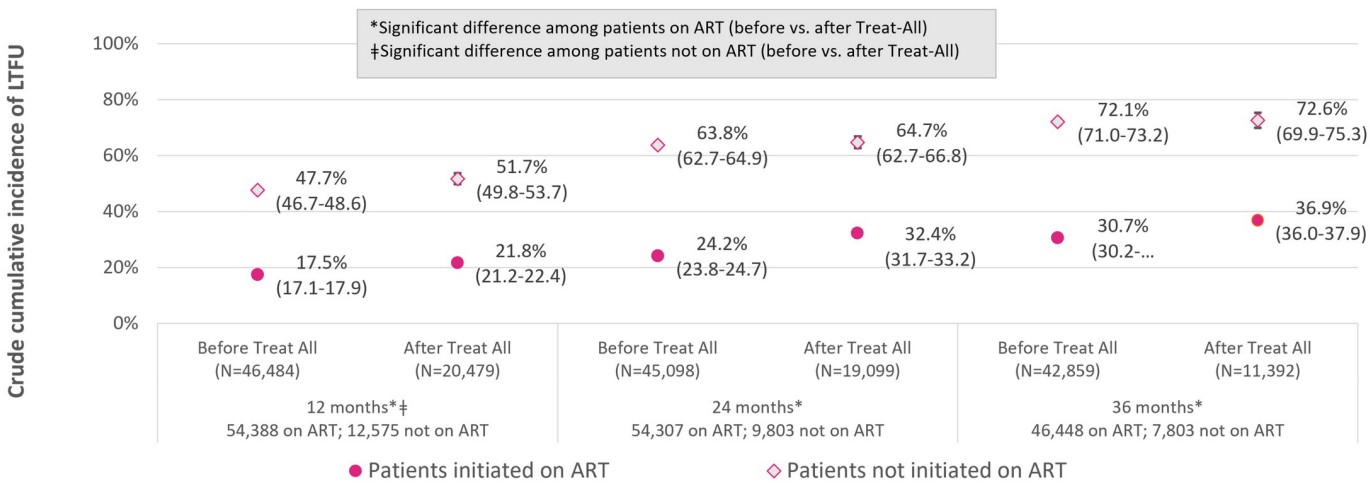

**Fig 5. Crude CI (95% CIs) of LTC at 12, 24, and 36 months after enrollment before vs. after adoption of universal HIV treatment guidelines, by treatment initiation status (on ART vs. not on ART) at ascertainment time point.** ART, antiretroviral therapy; CI, cumulative incidence; LTC, lost to clinic; 95% CI, 95% confidence interval.

**Table 3. Risks and hazard ratios of LTC associated with national adoption of universal HIV treatment guidelines.**

| Care outcome | Enrollment before guideline adoption* N (%) | Enrollment after guideline adoption N (%) | HR (95% CI) | aHR[†§] (95% CI; p-value) |
|---|---|---|---|---|
| **LTC** | | | | |
| 12 months after enrollment[♦] | 11,056 (23.8) | 5,135 (25.1) | 1.08 (0.96, 1.21) | 1.04 (0.94, 1.15); p = 0.454 |
| 24 months after enrollment[‡] | 13,997 (31.0) | 6,823 (35.9) | 1.19 (1.05, 1.36) | 1.13 (0.99, 1.27); p = 0.056 |
| 36 months after enrollment** | 15,946 (37.2) | 4,580 (40.2) | 1.11 (0.98, 1.27) | 1.11 (0.98, 1.25); p = 0.100 |
| **LTC among patients on ART before end of follow-up** | | | | |
| 12 months after enrollment | 6,271 (17.2) | 3,825 (21.3) | 1.30 (1.12, 1.51) | 1.25 (1.08, 1.44); p = 0.003 |
| 24 months after enrollment | 9,038 (24.2) | 5,510 (32.4) | 1.44 (1.25, 1.67) | 1.38 (1.19, 1.59); p < 0.001 |
| 36 months after enrollment | 11,073 (30.7) | 3,823 (36.9) | 1.30 (1.13, 1.50) | 1.34 (1.18, 1.53); p < 0.001 |
| **LTC among patients not on ART before end of follow-up** | | | | |
| 12 months after enrollment | 4,785 (47.7) | 1,310 (51.7) | 1.17 (1.00, 1.36) | 1.16 (0.96, 1.39); p = 0.117 |
| 24 months after enrollment | 4,959 (63.8) | 1,313 (64.7) | 1.03 (0.88, 1.19) | 0.99 (0.86, 1.15); p = 0.934 |
| 36 months after enrollment | 4,873 (72.1) | 757 (72.6) | 1.01 (0.87, 1.16) | 1.03 (0.90, 1.18); p = 0.656 |

aHR, adjusted hazards ratio; ART, antiretroviral therapy; HR, hazards ratio; LTC, lost to clinic.

*Reference group: Patients enrolling in care before adoption of universal HIV treatment guidelines.

[†]Adjusted for sex, age group, enrollment CD4, facility type, clinic location, and country income level.

[§]Transfer and death treated as competing events.

[♦]p-Value for interaction term between enrollment period and ART status by censoring time point: 0.400.

[‡]p-Value for interaction term between enrollment period and ART status by censoring time point: 0.002.

**p-Value for interaction term between enrollment period and ART status by censoring time point: 0.027.

patients not on ART by 12, 24, and 36 months after enrollment, the hazards of LTC did not differ by period of enrollment.

Table 4 presents the proportion of patients retained in care at clinic of enrollment at 12, 24, and 36 months after ART initiation, having a record of VL monitoring and having VS at each time point, along with crude and adjusted relative risks (aRRs) of each outcome by period of enrollment. Adjusting for patient and clinic characteristics, those enrolling in care after adoption of universal treatment guidelines were less likely to be retained in care at 12 months (aRR 0.95 [95% CI: 0.93, 0.98]; $p = 0.001$); 24 months (aRR 0.88 [95% CI: 0.84, 0.94]; $p < 0.001$), and 36 months after ART initiation (aRR 0.87 [95% CI: 0.82, 0.92]; $p < 0.001$). Among patients retained after ART initiation, those enrolling in HIV care after adoption of universal treatment guidelines were more likely to have VL monitoring at 12 months after ART initiation (aRR 1.15 [95% CI: 1.05, 1.26]; $p = 0.004$) and less likely at 36 months (aRR 0.86 [95% CI: 0.80, 0.92]; $p < 0.001$), with no difference at 24 months. Among patients with VL monitoring at 12, 24, and 36 months after ART initiation, the likelihood of VS at 24 months was marginally, but not substantively, higher among patients enrolling in HIV care after guideline adoption (aRR 1.03 [95% CI: 1.01, 1.04]; $p = 0.001$), with no differences at 12 or 36 months.

**Table 4. Risks of HIV care outcomes after ART initiation associated with national adoption of universal HIV treatment guidelines.**

| Care outcome | Enrollment before guideline adoption* N (%) | Enrollment after guideline adoption N (%) | RR (95% CI) | aRR[†] (95% CI); *p*-value |
|---|---|---|---|---|
| **Retention in care** | | | | |
| 12 months after ART initiation | 29,483 (75.1) | 12,935 (70.5) | 0.94 (0.91, 0.97) | 0.95 (0.93, 0.98); *p* = 0.001 |
| 24 months after ART initiation | 24,785 (64.8) | 9,565 (55.8) | 0.86 (0.81, 0.91) | 0.88 (0.84, 0.94); *p* < 0.001 |
| 36 months after ART initiation | 20,025 (55.0) | 5,016 (48.2) | 0.88 (0.82, 0.93) | 0.87 (0.82, 0.92); *p* < 0.001 |
| **VL testing among patients initiating ART and retained in care** | | | | |
| 12 months after ART initiation | 18,804 (63.8) | 9,485 (73.3) | 1.15 (1.04, 1.28) | 1.15 (1.05, 1.26); *p* = 0.004 |
| 24 months after ART initiation | 17,413 (70.3) | 7,017 (73.4) | 1.04 (0.99, 1.10) | 1.03 (0.99, 1.07); *p* = 0.166 |
| 36 months after ART initiation | 14,328 (71.6) | 3,195 (63.7) | 0.89 (0.80, 0.99) | 0.86 (0.80, 0.92); *p* < 0.001 |
| **VS among those retained in care with VL testing** | | | | |
| 12 months after ART initiation | 14,106 (86.0) | 7,050 (86.2) | 1.00 (0.98, 1.03) | 1.01 (0.98, 1.04); *p* = 0.360 |
| 24 months after ART initiation | 15,184 (87.2) | 6,229 (88.8) | 1.02 (1.00, 1.03) | 1.03 (1.01, 1.04); *p* = 0.001 |
| 36 months after ART initiation | 12,721 (88.8) | 2,885 (90.3) | 1.02 (1.00, 1.03) | 1.01 (0.99, 1.03); *p* = 0.363 |

aRR, adjusted risk ratio; ART, antiretroviral therapy; RR, risk ratio; VL, viral load; VS, viral suppression; 95% CI, 95% confidence interval.

*Reference group: Patients enrolling in care before adoption of universal treatment guidelines.

[†]Adjusted for sex, age group, enrollment CD4, initial regimen type, clinic location, facility type, and country income level.

Sensitivity analyses restricted to patients enrolling in the 13 to 24 months before adoption of universal HIV treatment guidelines and the first year afterwards showed consistent results for LTC outcomes at 12 months after enrollment, as well as retention, VL monitoring, and VS at 12 months after ART initiation (S2 Table). Post hoc sensitivity analyses that excluded countries with late adoption of universal HIV treatment guidelines in 2017 and 2018, where 24- and 36-month outcomes windows could have partially coincided with service disruptions related to the COVID-19 pandemic, also yielded results consistent with our main analyses (S3 and S4 Tables).

## Discussion

Using real-world service delivery data from 109 clinics across 25 countries, our study found that patient retention at 12, 24, and 36 months after ART initiation decreased among patients enrolling in care after national adoption of universal HIV treatment guidelines. Additionally, while VS was high before and after guideline adoption among patients retained in care and on ART, there was little improvement in annual VL monitoring.

Prior experimental and observational studies examining HIV care outcomes under universal treatment guidelines have focused on outcomes within 12 months after treatment initiation [9–13,15,16] and have reported mixed results in terms of patient attrition and retention. To our knowledge, no other studies using real-world service delivery data have examined longer-term HIV care outcomes at 24 and 36 months after enrollment and after ART initiation in the era of universal HIV treatment.

In accordance with prior research [4,5,11], we found that patients enrolling in care under universal HIV treatment guidelines initiated ART more rapidly, with a lower proportion of patients having no record of ART initiation. While observed improvements in ART initiation are encouraging, our study raises concerns about the capacity of HIV programs to support engagement in care for some patients after ART initiation. Study findings also raise concerns about the quality of HIV care in the era of universal HIV treatment. WHO has recommended annual VL monitoring after ART initiation since 2013 [24]; however, VL monitoring remains suboptimal, with more than one-quarter of patients at each annual time point in our study having no record of VL testing. Among patients enrolling after national adoption of universal treatment guidelines, we found no improvement in VL testing at 24 months, and VL monitoring at 36 months decreased among patients enrolling after guideline adoption. Observed decreases in VL monitoring at 36 months are surprising because the subset of patients enrolling in HIV care after guideline adoption with at least 36 months of follow-up time comprised more patients from high-income countries with greater capacity for VL testing [25]. Additionally, while the COVID-19 pandemic is known to have disrupted HIV-related services in many settings [26,27], most patients in our study (>95%) were in countries adopting universal treatment guidelines in 2016 or earlier, meaning that VL monitoring at 36 months after ART initiation reflected prepandemic VL monitoring practices. In sensitivity analyses that excluded countries where outcomes at 36 months could have coincided with the COVID-19 pandemic findings were consistent with our main analyses. Accordingly, observed gaps in annual VL monitoring among retained patients—particularly with increasing time since ART initiation—reinforce concerns about the adequacy of resource allocation for elements of HIV care that are essential for identifying adherence challenges and drug resistance and for guiding timely regimen switching [28]. Additionally, as other research has shown that patient monitoring is positively associated with retention [29], gaps in VL monitoring may represent missed opportunities for motivating patients to remain engaged in care.

Study limitations include well-known limitations of observational studies for causal attribution, along with possible underascertainment of deaths and transfers to other sites of care in routine HIV service delivery data [30–32]. As tracing studies have reported rates of undocumented (i.e., "silent") transfer ranging from 4% to 54% among patients lost to follow-up [30–32], true LTC may be overestimated in our study, and deaths and transfers are likely underestimated [33]. Although WHO began recommending the decentralization of HIV care in LMICs in the mid- to late 2000s, well before the adoption of universal HIV treatment guidelines in these settings [34–36] and we used a conservative definition of LTC [37], it is possible that decentralization has accelerated with the rollout of universal treatment policies, resulting more undocumented transfers, particularly among patients at centralized or tertiary care sites who silently transfer to peripheral HIV care facilities [31,32].

While we were able to adjust for selected patient characteristics, there may be important unmeasured differences between patients enrolling in HIV care and initiating treatment before and after the adoption of universal treatment guidelines. Other analyses have shown that patients entering HIV care in the era of universal treatment initiate treatment more rapidly [4,5] and that the expansion of HIV treatment eligibility criteria has resulted in the treatment of patients with earlier stage HIV disease [38]. While earlier treatment initiation is associated with improved clinical outcomes and reduced onward HIV transmission, rapid initiation of treatment after enrollment in HIV care is also associated with lower retention in care [39,40], and qualitative research has suggested that distress and uncertainty about HIV diagnosis, concerns about stigma, fear of lifelong medication, and other patient-level factors may contribute to attrition and lower treatment adherence among those rapidly initiating treatment, particularly among patients with early stage disease who do not feel unwell [41,42]. Our observed

decreases in retention after ART initiation in the era of universal treatment may reflect the fact that some patients, who would have been LTC prior to ART initiation before guideline adoption, initiated ART more rapidly.

We had limited data on patient characteristics and note that substantial decreases in CD4 testing after the adoption of universal treatment guidelines [43] constrain our ability to adjust for patient immunological status at the time of care entry and treatment initiation, which may be a source of bias in the estimates we report. We also were unable to adjust for pregnancy status or examine whether longer-term HIV care outcomes among pregnant women—a group that was eligible for immediate treatment and life-long ART prior to the adoption of universal HIV treatment guidelines—differed from those of other patients. Additionally, while we adjusted for clinic characteristics, such as location, facility type, and country income level, there may be important time-varying contextual and health system factors, including supply-side constraints related to increased demand for HIV treatment, which we were unable to adjust for in our analyses. It is noteworthy that for patients in our study who enrolled in HIV care and initiated ART before the adoption of universal HIV treatment guidelines, outcomes at 24 and 36 months occurred after guidelines had changed. While it is unlikely that routine follow-up care of patients established on ART differed by the timing of treatment initiation, practices related to ART readiness and adherence counseling may have differed substantially before and after the adoption of universal HIV treatment guidelines but we were unable to examine or control for such differences.

A further limitation is the nonrepresentativeness of IeDEA sites within countries and regions included in this analysis; almost half of our sample were in care at university and tertiary referral hospitals, and these sites are likely better resourced than other HIV clinics within the same geographic areas. Accordingly, our estimates of VL monitoring before and after the adoption of universal HIV treatment guidelines may be biased upwards. While consistent with other research [44,45], our estimates of VS may also be biased upwards, particularly as those without VL monitoring likely include patients disengaged from care—patients known to have higher rates of viral nonsuppression [46,47]. Additionally, our study findings may not be generalizable to other locations and contexts that were not included in our study. It is also possible that patients enrolling in care and initiating treatment during the first year after the adoption of universal HIV treatment guidelines are not representative of those enrolling during subsequent years, with HIV outcomes improving among patients enrolling 2 to 3 years after the initial period of guideline adoption.

Key strengths of our study include the use of real-world service delivery data from a large sample to examine longer-term programmatic outcomes after the adoption of universal HIV treatment guidelines across diverse country settings. Associations observed in our heterogeneous sample of patients from diverse settings across multiple regions and different years of guideline adoption may be broadly reflective of HIV care outcomes among patients who were ART naïve at enrollment in HIV care during the years surrounding the adoption of universal HIV treatment guidelines in these settings.

While the adoption of universal HIV treatment guidelines has expanded access to and uptake of timely treatment for PLWH, our findings raise concerns about existing service delivery strategies and capacity for longer-term retention of patients in the era of universal treatment. Our findings of increased LTC at 24 and 36 months after enrollment and decreased retention at all time points after ART initiation suggest a risk of worsened patient outcomes. Additionally, while VS rates at all time points are high, our study highlights suboptimal VL monitoring, particularly with increased duration of time in care. Because of lags in the availability of data extracted from patient records and databases, we were only able to examine outcomes among patients enrolling in HIV care and initiating treatment during the first year after

the adoption of universal treatment guidelines. However, our findings underscore the critical importance of monitoring long-term HIV care outcomes as additional data become available, as well as examining HIV care outcomes among groups known to be at increased risk of attrition and poor VS, including pediatric patients and pregnant and postpartum women. Equally vital are efforts to identify and address health system, community and patient-level determinants of attrition before and after ART initiation, and barriers to adherence and viral nonsuppression in the era of universal treatment of all PLWH.

## Supporting information

**S1 Checklist. STROBE Statement.**
(DOCX)

**S1 Text. Acknowledgments.**
(DOCX)

**S1 Concept Proposal. Concept sheet: Multiregional analysis.**
(PDF)

**S1 Table. Baseline characteristics among patients enrolling in care at least 24 and 36 months before database closure.**
(DOCX)

**S2 Table. Sensitivity analyses restricted to patients enrolling 13–24 months before and 12 months after national adoption of universal HIV treatment guidelines.**
(DOCX)

**S3 Table. Risks and hazards of LTC associated with national adoption of universal HIV treatment guidelines in countries introducing guideline changes before 2017.**
(DOCX)

**S4 Table. Relative risks of HIV care outcomes after ART initiation associated with national adoption of universal HIV treatment guidelines in countries introducing guideline changes before 2017.**
(DOCX)

## Acknowledgments

We sincerely thank the staff at contributing sites, as well as IeDEA regional data managers. The full list of Acknowledgments can be found in the Supporting information (S1 Text).

## Author Contributions

**Conceptualization:** Olga Tymejczyk, Denis Nash.

**Data curation:** Kara Wools-Kaloustian, Awachana Jiamsakul, Marco Tulio Luque Torres, Jennifer S. Lee, Lisa Abuogi, Vohith Khol, Fernando Mejía Cordero, Keri N. Althoff, Matthew G. Law.

**Formal analysis:** Ellen Brazier.

**Funding acquisition:** Kara Wools-Kaloustian, Keri N. Althoff, Matthew G. Law, Denis Nash.

**Methodology:** Ellen Brazier, Olga Tymejczyk, Denis Nash.

**Writing – original draft:** Ellen Brazier.

**Writing – review & editing:** Ellen Brazier, Olga Tymejczyk, Kara Wools-Kaloustian, Awachana Jiamsakul, Marco Tulio Luque Torres, Jennifer S. Lee, Lisa Abuogi, Vohith Khol, Fernando Mejía Cordero, Keri N. Althoff, Matthew G. Law, Denis Nash.

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
