## [Editor Report · Decision Letter 0]

8 Aug 2023

Dear Dr Brazier, 

Thank you for submitting your manuscript entitled "Long-term HIV care outcomes under “Treat-All” guidelines: A retrospective cohort study" for consideration by PLOS Medicine.

Your manuscript has now been evaluated by the PLOS Medicine editorial staff as well as by an academic editor with relevant expertise and I am writing to let you know that we would like to send your submission out for external peer review.

Please re-submit your manuscript within two working days, i.e. by Aug 10 2023 11:59PM.

Kind regards,

Louise Gaynor-Brook, MBBS PhD

Senior Editor

PLOS Medicine

---

## [Decision Letter · Decision Letter 1]

22 Nov 2023

Dear Dr. Brazier,

Thank you very much for submitting your manuscript "Long-term HIV care outcomes under “Treat-All” guidelines: A retrospective cohort study" (PMEDICINE-D-23-02140R1) for consideration at PLOS Medicine. 

Your paper was evaluated by a senior editor and discussed among all the editors here. It was also discussed with an academic editor with relevant expertise, and sent to three independent reviewers, including a statistical reviewer. The reviews are appended at the bottom of this email and any accompanying reviewer attachments can be seen via the link below:

[LINK]

Whilst we will not be able to accept the manuscript for publication in the journal in its current form, we would like to consider a revised version that fully addresses the reviewers' and editors' comments. Obviously we cannot make any decision about publication until we have seen the revised manuscript and your response, and we plan to seek re-review by one or more of the reviewers. 

We expect to receive your revised manuscript by Dec 13 2023 11:59PM. Please email me (lgaynor@plos.org) if you have any questions or concerns.

We look forward to receiving your revised manuscript. 

Sincerely,

Louise Gaynor-Brook, MBBS PhD

lgaynor@plos.org

plosmedicine.org

Thank you for your patience with a longer assessment process than we anticipated, and apologies for the delay in providing you with an editorial decision. 

Comments from the Academic Editor: 

Consideration of the national settings and programmes is required, which may underlie some of the results noted and may need to be addressed either by additional analysis and/or points in the Discussion.

The authors need to indicate and justify why there were no Southern Africa/ West Africa data included - those IeDEA data exist and most of HIV infected people live in Southern Africa. This point needs to be addressed as a limitation in the Discussion, and the text needs to be very clear of the settings from which the data come (perhaps even in the Title).

Wealth Index is not a sufficient covariate to indicate differences between countries. Instead there needs to be a more insightful variable that indicates something about access to health care/ access to ART/ or possibly early versus late TA policy adopters - anything that says something about the public health system would be especially important for the countries in Africa.

There seems to be some confusion regarding being in HIV care and being on ART - is the start of the follow up from accessing HIV care eg for testing, or from starting ART, or both? Being in HIV care may say a lot about the individual whereas ART initiation says a lot about the system with adherence then saying something about the individual too.

ART status is taken at censoring points but this then is likely to be strongly correlated with being in care/LTC? Instead, it would be better to allow for timing of ART initiation during the follow up period, or if not available, whether ART was already initiated at the preceding time point (whether that was a clinic visit 3 or 6 months early or the preceding censoring point).

People who started ART before the national TA policy was introduced are different from those starting ART after TA policy - and that is not sufficiently addressed. In particular it would be likely that those initiating before TA already have some positive link with the clinic while after TA, they may have attended a particular clinic for HIV testing (and if positive would have been initiated on ART) but then moved away and access ART elsewhere, as the initial clinic would not be the most convenient for long-term care. This needs to be addressed.

Before TA policies were introduced, pregnant women would already have been HIV tested and if positive would have been initiated on ART for life - this policy was common globally. The health care system would be geared towards those women and their children, even long-term. How would this have been changed by a TA policy? Can this be addressed?

Of course, what is most important from a public health perspective is to retain those who positively want to be on ART in long term care, in a clinic (or within a referral system that allows any move to be visible) that is convenient long term for the individual concerned. The results noted on page 17 are relevant to this point and this point should be explored in the discussion.

General comments:

Please include line numbers in your revised manuscript, ideally not starting from 1 with each new page.

Please use person-first language throughout e.g. “patients who are ART-naïve” rather than “ART-naïve patients”

Throughout the paper, please adapt reference call-outs to the following style: "... every year [1,2]." (noting the absence of spaces within the square brackets).

Title: Please revise your title according to PLOS Medicine's style. We suggest “Long-term HIV care outcomes under World Health Organization “Treat-All” guidelines: A retrospective cohort study in 25 countries” or similar

Abstract:

Please structure your abstract using the PLOS Medicine headings (Background, Methods and Findings, Conclusions), combining the Methods and Findings sections into one section.

Please note that up to 500 words are permitted.

Abstract Background: Provide expand upon the context of why the study is important. The final sentence should clearly state the study question.

Abstract Methods and Findings:

Please provide the abbreviation IeDEA 

Please clarify what is meant by 'database closure’

Please provide brief demographic details of the study population (e.g. sex, age, ethnicity, etc)

Please include the study design, population and settings (e.g. regions included in IeDEA), years during which data were collected, and length of follow up (eg, in mean, SD, and range).

Please define CI and aHR at first use. 

Please provide p values alongside 95% CIs where available. 

Please specify ‘months’ after 12 and 24 when discussing hazards of LTC

Please include the important dependent variables that are adjusted for in the analyses.

In the last sentence of the Abstract Methods and Findings section, please describe 2-3 of the main limitations of the study's methodology.

Abstract Conclusions:

Please begin your Abstract Conclusions with "In this study, we observed ..." or similar, to summarize the main findings from your study, without overstating your conclusions. Please emphasize what is new and address the implications of your study, being careful to avoid assertions of primacy. 

Author Summary:

In the final bullet point of ‘What Do These Findings Mean?’, please describe the main limitations of the study in non-technical language.

Introduction:

Please expand a little on the background to your study and address other past research. 

Methods:

Did your study have a prospective protocol or analysis plan? Please state this (either way) early in the Methods section. If a prospective analysis plan was used in designing the study, please include the relevant prospectively written document with your revised manuscript as a Supporting Information file to be published alongside your study, and cite it in the Methods section. If no such document exists, please make sure that the Methods section transparently describes when analyses were planned, and if/when reported analyses differed from those that were planned. Changes in the analysis-- including those made in response to peer review comments-- should be identified as such in the Methods section of the paper, with rationale. If a reported analysis was performed based on an interesting but unanticipated pattern in the data, please be clear that the analysis was data-driven.

Please define NRTI+NNRTI at first use,

Please provide the names of the institutional review boards that provided ethical approval.

Please ensure that the study is reported according to the STROBE guideline, and include the completed STROBE checklist as Supporting Information. Please add the following statement, or similar, to the Methods: "This study is reported as per the Strengthening the Reporting of Observational Studies in Epidemiology (STROBE) guideline (S1 Checklist)." The STROBE guideline can be found here: http://www.equator-network.org/reporting-guidelines/strobe/ When completing the checklist, please use section and paragraph numbers, rather than page numbers which will likely no longer correspond to the appropriate sections after copy-editing.

Please report the years during which data used in this study were collected.

Results: 

Please define IQR at first use.

In the first, third and fifth sentences on page 7 , please specify that the results presented apply to all patients in the cohort (i.e. not defined by before/after Treat All)

Please quantify the results presented in the main text, providing 95% CIs and exact p values. 

Please indicate which factors are adjusted for the main text relating to Table 5.

Where aHR/aRR are provided, please ensure to specify the comparison group.

Please define the length of follow up (eg, in mean, SD, and range).

Discussion:

Please present and organize the Discussion as follows: a short, clear summary of the article's findings; what the study adds to existing research and where and why the results may differ from previous research; strengths and limitations of the study; implications and next steps for research, clinical practice, and/or public policy; one-paragraph conclusion.

Please remove the Conclusions subheading within your Discussion 

Please remove the information on competing interests, funding and data sharing from the

end of the main text. In the event of publication, this information will appear in the article

metadata, via entries in the submission form.

Figures:

Please define all abbreviations used in the figure legend of each figure.

Tables:

Please define all abbreviations used in the table legend of each table, including in the supplementary files.

When a p value is given, please specify the statistical test used to determine it in the legend.

Please provide exact p values, rather than e.g. <.001

References:

Please ensure that journal name abbreviations match those found in the National Center for Biotechnology Information (NCBI) databases (http://www.ncbi.nlm.nih.gov/nlmcatalog/journals), and are appropriately formatted and capitalised. Six authors should be listed prior to ‘et al’. Please also see https://journals.plos.org/plosmedicine/s/submission-guidelines#loc-references for further details on reference formatting. 

Comments from the reviewers:

Reviewer #1: In this paper, the authors present an analysis of data from the IeDEA consortium, aiming to answer the question of whether people with HIV had better follow-up outcomes following the introduction of Treat-All guidelines. While this analysis does account for the competing risk of death when assessing the effect of the Treat-All guidelines on the outcome of care retention, there are several aspects of this analysis that require clarification and further justification, as described in my comments below.

1. Please complete and include the STROBE checklist (see https://www.equator-network.org/reporting-guidelines/strobe/)

2. Abstract: The regions that the 25 countries were drawn from and the period of time that the data corresponds to should be mentioned. 

3. Methods:

a. What statistical program was used to analyse the data? If R, please be sure to cite the relevant packages. 

b. It is stated that multivariable cause-specific hazards regression was applied - but what model specifically was assumed? How was clustering within clinic accounted for in the cause-specific hazards regression models? Please provide details.

c. What is meant by "correlates of LTC"? What research question is this analysis aiming to answer? Generally speaking, if interest is in the effect of a covariate or an exposure on an outcome, the researcher must consider all variables that may act as confounders for the relationship between that covariate/exposure on the outcome, and adjust for these in an analysis. Thus, separate analyses may be required for each exposure/covariate and outcome pair. Tables of regression coefficients or hazard ratios as presented in Table 4 are very difficult to interpret, and often do not have a valid causal interpretation and thus should not be included. Please see Westreich D and Greenland S, The Table 2 Fallacy: presenting and interpreting confounder and modifier coefficients. American Journal of Epidemiology, 2013;177:92-298. (https://doi.org/10.1093/aje/kws412) 

d. I would think that those patients with ART initiation are likely to have been treated for longer at a clinic than those patients without ART initiation, and thus would have a lower incidence of loss to follow up. This does not appear to have been commented on: it has implications for interpretation of results. The analysis of patients who commenced ART during their follow up is conditioning on a future event; a more appropriate analysis would treat ART initiation as a time-varying exposure. Further, after the Treat-All guidelines were introduced I can imagine that more patients commenced ART earlier; thus the effect of the introduction of the guidelines in those patients who commenced ART cannot be disentangled from the differences between the group of patients who had commenced ART before the introduction of Treat-All and the group who commenced ART after Treat-All. The approach taken requires justification or deletion. 

e. Patients who were enrolled in clinics prior to the date of adoption in their country will, depending on their date of enrolment, still be enrolled after the date of adoption. Thus patients could have a time-varying exposure to the Treat-All policy. How was this accounted for in the analysis? What are the impacts of this on conclusions?

4. Results

a. How complete was the dataset? Were all patient-level characteristics available for all patients in the dataset, or was the analysis restricted to those patients with complete data? Missing CD4 count at enrolment is mentioned, but what about other characteristics? If there was missingness in other characteristics, what implications might this have for the results and their interpretation?

b. In several places on Page 7, results relating to proportions of patients experiencing outcomes are mentioned without any accompanying proportions or indications of where readers may find these. For example, in the first paragraph: where are the proportions of patients with transfers at each of the time points mentioned in the second sentence?

c. The modern usage of p-values favours a more holistic interpretation of results, rather than simply assessing whether a p-value happens to be small. When a large amount of data is analysed, small differences can have very small p-values: statistical significance may not equate to statistical significance. The American Statistical Association's statement on p-values, available at https://doi.org/10.1080/00031305.2016.1154108 provides further context and guidance. An assessment of the differences between groups guided by clinical rather than statistical significance, with consideration of variability, would be far more appropriate here. 

d. As indicated in my comments on the Statistical Analysis section, it is extremely difficult to interpret the results presented in Table 4 in a useful way. Thus this table and associated discussion should be deleted. 

5. Discussion

a. For those countries with national adoption dates occurring in 2017-2018, the disruption caused by the COVID-19 pandemic and associated restrictions could have had a large impact on outcomes at 24 to 36 months. Although this is mentioned in the discussion, a sensitivity analysis where data from 2017 and 2018 is excluded is required. 

b. Given my concerns above regarding the analysis of the patients who had initiated ART, the conclusions regarding decreased retention after ART initiation need to be further justified.

Reviewer #2: This is a large, retrospective cohort analysis by the IeDEAL group, using data from 109 clinics in 25 countries to study the impact of the Treat-All (TA) policy on retention on ART, viral load (VL) coverage and VL suppression. The manuscript is very well written, the complex methodology has been explained clearly and the results are nicely displayed in graphs and tables. There are no ethical issues.

Major points

1. The main benefit of TA is higher uptake of ART and prevention of complications through earlier treatment. While uptake of ART among persons diagnosed and enrolled in care will be higher after TA, this benefit could be reduced or negated by higher attrition from care. In other words, of persons enrolled in care, the net percentage of those who started ART and were then retained on ART at the 3 time points, should be compared between the 2 time periods. Does the study provide this insight? The authors have stratified LTC between those starting ART and not starting ART, and compared retention among those who started ART, but this is not the same. Can the authors explain?

2. All before/after analyses have challenges with confounding by impacts that vary over time. This is also relevant in this study with its broad geographical scope and variable calendar time of the intervention. I believe that the authors should pay more attention to this in the Discussion. How do the authors assess the impact of external factors, including the C19 pandemic, the transition to INSTIs and possible other changes that may have happened in tandem with the TA policy, on the findings of their study?

3. The authors state that "our heterogeneous sample of patients from diverse settings across multiple regions and different years of Treat-All adoption" is a strength of the study. However, it seems difficult to compare the effects of the transition to TA in the USA to those in Africa several years later, with the vastly different health systems and populations. In my view, adjustment for income level of the country does not fully address this. For instance, readers may be concerned that the overall outcomes and conclusions could be less relevant for Africa due to being driven by results from North America and Asia. Could the authors elaborate on this?

4. Enrolment of the post TA cohort was directly after the introduction of the change in policy. Could disruptions in health systems due to the change in policy have a temporary effect on the study outcomes? An example of this is supply challenges due to increased demand. If so, participants enrolled in the period immediately after the TA policy introduction may not be representative of individuals started on ART later. 

Reviewer #3: This paper reports a comprehensive analysis of retrospective data on retention, viral suppression and other outcomes in a large multi-national cohort of patients enrolling in HIV care. The data come from the IdEA consortium and include information on nearly 67,000 patients from 109 clinics in 25 countries. The main objective of the analyses is to compare outcomes in patients enrolling before and after national guidelines changed to a Treat-All policy in each country.

This is a very well written paper. The rationale, methods and results are generally clearly presented and there is a good discussion of the policy implications of the findings. Strengths of the study include the availability of service delivery data from a very large sample of patients in a wide range of settings, with follow-up to 36 months after enrolment in HIV care. Previous studies have mostly been based on more restricted geographical settings and shorter follow-up periods, so this study makes an important contribution to the field. The findings of lower retention and little improvement in viral load monitoring in patients enrolling after the change to Treat-All have clear implications for policy and practice.

An important limitation is that there are no study sites in Southern Africa, where a large proportion of HIV patients live, although this reflects the coverage of the IdEA consortium, over which the authors have little control. Furthermore the study only includes data on patients enrolling in care during the first year after the guideline change. It usually takes time for guidelines to "bed in" and for health services to adapt to the new context, and especially the challenge of serving a larger number of patients on ART. It will be important to carry out further analyses once the new guideline has been in place for a longer time. These points are not discussed.

Apart from these observations, I only had a few comments which are listed below.

1. Some readers may be confused by the fact that WHO made the recommendations for Treat-All in 2015 and yet some sites included in the study changed to Treat-All as early as 2012. A brief comment on the reasons for this might be helpful.

2. A key issue is that this is an observational analysis, and that the groups of patients enrolling before and after the guideline change may differ in ways that would affect the outcomes being measured, leading to selection bias. Table 1 shows that there are few differences in basic demographic and clinical variables, and the effect measures are adjusted for these variables. However, there may be differences in other unmeasured variables. For example, a Treat-All policy might encourage a wider range of patients to present for care, possibly including a higher proportion of those with intrinsically lower health-seeking behavior, and consequently lower retention and/or adherence to treatment. I note that hazard ratios are considerably attenuated after the limited adjustment that is possible, and might be further attenuated if it were possible to adjust for additional variables. This is mainly a point for discussion.

3. Some of the analyses are stratified by whether or not patients had initiated ART before end of follow-up. But it is not clear how long they took to initiate ART, so how much of their follow-up period was while they were on ART. We are told that 87.6% of those enrolling in HIV care after the guideline change were on ART by 12 months, but some more information on the delay in initiating ART might be helpful.

4. A smaller proportion of patients enrolling after the guideline change had CD4 counts at baseline, which is not surprising as ART initiation was no longer dependent on the CD4 count value. However, this lack of data compromises the ability to control for an important clinical variable. Again, this is mainly a point for discussion.

5. Figs 3 and 4: The legend for these figures needs to explain that 95% CIs are shown and also define what is meant by "significant". Also, it would be possible to display the confidence intervals on the diagrams.

6. Table 3 does NOT show the "hazards" of LTC at each time-point. The first two columns show (I think) the cumulative incidence or risk of LTC by each time-point. It is in columns 3 and 4 I think that hazard ratios are shown. Could this be clarified please?

7. Table 4: A minor point, but with a covariate with several categories (e.g. age) it would be usual to show an LRT p-value for the overall evidence of any effect of that variable on the outcome, in addition to the effects and CIs for the individual categories.

8. Supplement Table 1: Please check the footnotes, which should define both HR and RR as these are both shown in the table.

[LINK]

---

## [Decision Letter · Decision Letter 2]

19 Jan 2024

Dear Dr. Brazier,

Thank you very much for re-submitting your manuscript "Long-term HIV care outcomes under universal HIV treatment guidelines: A retrospective cohort study in 25 countries" (PMEDICINE-D-23-02140R2) for consideration at PLOS Medicine.

I have discussed the paper with our academic editor and it was also seen again by three reviewers. I am pleased to tell you that, provided the remaining editorial and production issues are fully dealt with, we expect to be able to accept the paper for publication in the journal.

[LINK]

Please let me know if you have any questions, and we look forward to receiving the revised manuscript.   

Sincerely,

Richard Turner PhD, for Louise Gaynor-Brook, MBBS PhD

Consulting Editor, PLOS Medicine

plosmedicine@plos.org

Requests from Editors:

In the abstract and main text, we ask you to quote p values alongside 95% CI, where available. 

At line 38 (abstract), we suggest "More than half ..." or similar. 

At lines 85-86, we ask you to remove "25%" and "more than 30%", bearing in mind that these are relative increases. You may quote the relevant HR and CI if you wish. 

We ask you to list the 25 countries, e.g., early in the Results section (main text). 

At line 468 we suggest "While the adoption ... has ...". 

Noting information in the tables, please quote exact p values or "p<0.001" for the smaller values. 

In the reference list, please use the journal name abbreviation "PLoS ONE". 

Reference 3 may be missing page numbers or a PII: please add additional information as needed. 

To reference 18 and any other preprints, please add "[preprint]", or update the citation with the relevant published article. 

Comments from academic editor:

My comment regarding pregnant women and the provision of universal ART for this group from first antenatal visit which has been available for 2 decades (and from before Universal treatment guidelines for all became available) is not really covered by the sentence in the final paragraph of the discussion. This is not an issue of 'crowding out' but a concern as to how changes in this group contribute to the findings. This would be most likely in the African and South American context - but it would be nice to see whether there is any evidence of retention in care in this group was improved after treatment for all policy became the norm. I do not know if pregnancy status data were available in this dataset.

Comments from Reviewers:

*** Reviewer #1: 

I thank the authors for their responses to my comments on the previous version of this manuscript. I have a few follow-up questions and comments.

1. (This comment relates to the authors' response to my comment 3(b) on the previous version.) Please state in the Methods section that clustering of patients within clinics was accounted for when fitting the Cox proportional hazards models through the use of a robust sandwich estimator for the covariance matrix. 

2. Were the assumptions of proportional hazards valid here? 

3. In Table 1, rather than (or at least in addition to) including p-values to assess differences between the characteristics of patients enrolled before and after guideline adoption, I would recommend the calculation of standardised differences - these do not suffer from the issue of small difference + large sample size issue of p-values, and instead give some idea of the magnitude of the differences between groups. 

4. In Table 1, rather than 95% confidence intervals for the mean time in days to ART initiation, reporting means and standard deviations and medians and lower and upper quartiles would be more informative, since characteristics of the sample are being described.

5. The issue of the interpretation of p-values raised in point 4(c) of my review of the previous version of this manuscript does not appear to have been thoroughly addressed. For example, on page 9: lines 325-326 ("small, but significant"); line 332 ("significantly higher"); line 347 ("but differences did not reach statistical significance"). I recommend that these sentences be re-worded in terms of clinical significance of observed differences and the range of values supported by the data through the confidence intervals for estimates. 

6. In Table 3, rather than noting which care outcomes had interaction terms with p-values below some arbitrary cut-off, please provide all interaction term p-values.

*** Reviewer #2: 

My comments and the extensive, very good comments from colleague reviewers, including those related to statistical issues, have been addressed well by the authors

Important limitations of the study remain but these have been acknowledged clearly in the Discussion

The conclusions are sufficiently supported by the data and are highly relevant for HIV care and treatment programs

In my view, the manuscript is ready for publication in PLoS Medicine

*** Reviewer #3: 

I have carefully reviewed the revised version of the manuscript as well as the authors' responses to the reviews and editorial comments. I would like to commend the authors for the care with which they have revised the paper. I am very satisfied with the changes they have made in response to my comments. I only have the following final comments and suggestions but do not need to see the paper again:

1. I note that a suggestion was made to the authors to report more p-values for comparisons in the Abstract or Text. However, I do not agree with this suggestion. I take the same view as the authors, that while there is continuing debate on this issue, current best practice is to de-emphasize the reporting of p-values, especially in studies (like this one) where very large sample sizes mean that even the smallest observed differences are deemed "statistically significant". As the authors note, the Confidence Intervals provide much more useful information about the size and precision of the estimates of interest.

2. I commented previously that the term "hazard" is often used in the paper when what is meant is really "hazard ratio" - that is, an effect estimate (comparing different exposure groups). However, there are still instances (e.g. Line 344 and Table 3) where greater clarity would be achieved by using the term "hazard ratio".

3. The lower proportion of patients with a viral load measure in the group enrolled after the guideline change is a surprising and concerning result that conflicts with the findings at 12m and 24m. I note that a much smaller number of patients in this group have data available at 36m, and suspect that they may be biased towards those enrolled at earlier time periods. If this is so, and if completeness of VL monitoring has been increasing over time, might this be an explanation for these contrasting results?

***

[LINK]

---

## [Decision Letter · Decision Letter 3]

22 Feb 2024

Dear Dr Brazier, 

On behalf of my colleagues and the Academic Editor, Marie-Louise Newell, I am pleased to inform you that we have agreed to publish your manuscript "Long-term HIV care outcomes under universal HIV treatment guidelines: A retrospective cohort study in 25 countries" (PMEDICINE-D-23-02140R3) in PLOS Medicine.

I appreciate your thorough responses to the reviewers' and editors' comments throughout the editorial process. We look forward to publishing your manuscript, and editorially there are only a few remaining minor stylistic/presentation points that should be addressed prior to publication. We will carefully check whether the changes have been made. If you have any questions or concerns regarding these final requests, please feel free to contact me at aschaefer@plos.org.

Please see below the minor points that we request you respond to:

1) l.39: Please report the interquartile range together with the median (“median age was 34 years.”).

2) Please throughout the main text (including the abstract), report statistical information as follows to improve clarity for the reader “22% (95% CI [13%,28%]; p</=)”. Please note the use of commas to separate upper and lower bounds, as opposed to hyphens as these can be confused with reporting of negative values.

3) We understand that in the last revision you were given the option to include the data in the Author Summary, but after further discussion with the editorial team, we feel that the Author Summary should be understandable to the lay reader and should not include data or technical language/jargon. We apologize for any inconvenience this may have caused. Editorial suggestion: Compared with patients enrolling in HIV care and initiating HIV treatment before national adoption of universal treatment guidelines, those enrolling and initiating treatment after guideline adoption had a higher chance of LTC at 12 months, 24 months and 36 months after enrollment.

3) Please check again carefully for the appropriate use of "hazard ratio" instead of "hazard" (e.g. l.44/l.86/l.351-360).

4) In the references, please replace the word “cited” with “accessed”.

5) Please add the following statement, or similar, to the Methods: "This study is reported as per the Strengthening the Reporting of Observational Studies in Epidemiology (STROBE) guideline (S1 Checklist)."

6) To help us extend the reach of your research, please provide any X (formerly known as Twitter) handle(s) that would be appropriate to tag, including your own, your co-authors’, your institution, funder, or lab. Please enter in the submission form any handles you wish to be included when we post about this paper.

PRESS

Sincerely, 

Alexandra Schaefer, PhD

On behalf of:

Louise Gaynor-Brook, MBBS PhD 

Senior Editor 

PLOS Medicine